# Physiological Aspects of World Elite Competitive German Winter Sport Athletes

**DOI:** 10.3390/ijerph19095620

**Published:** 2022-05-05

**Authors:** Paul Zimmermann, Jan Wüstenfeld, Lukas Zimmermann, Volker Schöffl, Isabelle Schöffl

**Affiliations:** 1Department of Cardiology, Klinikum Bamberg, 96049 Bamberg, Germany; 2Interdisciplinary Center of Sportsmedicine Bamberg, Klinikum Bamberg, 96049 Bamberg, Germany; luki.z@web.de (L.Z.); volker.schoeffl@me.com (V.S.); isabelle.schoeffl@me.com (I.S.); 3Division of Exercise Physiology and Metabolism, Department of Sport Science, University of Bayreuth, 95447 Bayreuth, Germany; 4Institut of Applied Training Science, Leipzig, 04109 Leipzig, Germany; janwuestenfeld@hotmail.com; 5Department of Orthopedic and Trauma Surgery, Klinikum Bamberg, 96049 Bamberg, Germany; 6Department of Orthopedic and Trauma Surgery, Friedrich-Alexander University Erlangen-Nurnberg, 91054 Erlangen, Germany; 7School of Clinical and Applied Sciences, Leeds Beckett University, Leeds LS1 3HE, UK; 8Section of Wilderness Medicine, Departement of Emergency Medicine, University of Colorado School of Medicine, Aurora, CO 80045, USA; 9Department of Pediatric Cardiology, Friedrich-Alexander-University Erlangen-Nurnberg, 91054 Erlangen, Germany

**Keywords:** cardiopulmonary exercise testing, ski-mountaineering, biathlon, Nordic-Cross Country, training, athlete physique

## Abstract

Nine Ski mountaineering (Ski-Mo), ten Nordic-Cross Country (NCC) and twelve world elite biathlon (Bia) athletes were evaluated for cardiopulmonary exercise test (CPET) performance as the primary aim of our descriptive preliminary report. A multicenter retrospective analysis of CPET data was performed in 31 elite winter sports athletes, which were obtained in 2021 during the annual medical examination. The matched data of the elite winter sports athletes (14 women, 17 male athletes, age: 18–32 years) were compared for different CPET parameters, and athlete’s physique data and sport-specific training schedules. All athletes showed, as estimated in elite winter sport athletes, excellent performance data in the CPET analyses. Significant differences were revealed for VE _VT2_ (respiratory minute volume at the second ventilatory threshold (VT2)), highest maximum respiratory minute volume (VE_maximum_), the indexed ventilatory oxygen uptake (VO_2_) at VT2 (VO_2_/kg _VT2_), the oxygen pulse at VT2, and the maximum oxygen pulse level between the three professional winter sports disciplines. This report provides new evidence that in different world elite winter sport professionals, significant differences in CPET parameters can be demonstrated, against the background of athlete’s physique as well as training control and frequency.

## 1. Introduction

Ski mountaineering (Ski-Mo), Nordic-Cross Country (NCC), and Biathlon (Bia) are known to be the most challenging winter sports because they involve whole-body movements, uphill locomotion, and altitude environments [1,2,3,4,5,6,7].

The different physiological demands and cardiopulmonary adaptions have been of interest in previous studies evaluating athlete’s performance by analyzing skiing techniques, training methods and echocardiographic parameters [7,8,9,10]. Therefore, individual athletic features and training schedules of these extreme endurance sports have been analyzed in previous studies for their pronounced structural and hemodynamic cardiac remodeling of the left heart as well as their cardiopulmonary exercise testing (CPET) performance [4,5,7,9,10,11]. Nevertheless, Ski-Mo is a rapidly growing winter sport, but despite its popularity, research on this discipline and its specific physiological demands are scarce [11,12].

The term athlete’s heart is defined as a sport-specific cardiac remodeling due to different structural, physiological, electro-physiological and functional sport-specific reasons for cardiac remodeling [13,14,15]. These sport-specific remodeling processes are evaluated with CPET analyses, cardiac imaging, and clinical cardiological assessment. Different training stimuli—represented by training schedule and training frequency—are pronounced factors for individually varying cardio-physiological adaption [7,15,16,17,18,19]. As the training schedule as well as the impact of training are comparably distributed between these three elite winter sports professionals, they represent the ideal comparison group for sport-specific cardiac remodeling.

In our preliminary report of these professional winter sport athletes, we elucidate the exercise-induced cardiopulmonary adaption by CPET. In winter sport, CPET is known as an important tool with which to analyze the individual athlete’s performance level and training status [20,21]. In recent decades, CPET has emerged as the main analyzing tool to manage peri-seasons training schedule levels and improve an athlete’s performance [4,8,20,22,23,24]. Nevertheless, there is a lack of data about interdisciplinary comparisons of world elite winter sport professionals’ CPET data and the individual physique status of individual athletes and training impact.

Therefore, the aim of our present descriptive preliminary reporting is to reveal physiological performance differences by utilizing CPET data in German elite competitive winter sports athletes.

## 2. Materials and Methods

### 2.1. Study Design

This was a multicenter retrospective analysis of CPET data in 31 elite winter sport athletes, which were obtained in 2021 during the annual medical sports medicine check-up. The matched data of the elite winter sports athletes (14 women, 17 male athletes, age: 18–32 years) were compared for different CPET parameters, athlete’s anthropometric data and sport-specific training aspects.

### 2.2. Participants

Thirty-one young elite winter sport professionals were included in the study population, all of whom were active members of the German National winter sport Team, participating in world championships as well as World cups, and they were examined during the season of 2021. All participants (14 women, 17 male athletes, age: 18–32 years) in this study had a comparable performance level in their sport discipline. All participants underwent an individual sports medicine check-up in their sports medicine performance center—Institut of Applied Training Science, Leipzig or Interdisciplinary Center of Sportsmedicine, Klinikum Bamberg in the preseason preparation. No participant was infected with Corona virus during the severe COVID-19 pandemic situation and so none had to be excluded from the annual performance testing. Male athletes had an age of 18–32 years, had a height between 175–186 cm, were of a weight between 65–81 kg and had a Body Mass Index (BMI) of 18–25 kg/m^2^. The participating female athletes were between 18–31 years, had a height of 154–176 cm, had a weight between 47–69 kg and showed a BMI of 18–23 kg/m^2^. All participants were professional athletes with a total amount of 20–25 training hours per week—based on the individual high volume training schedule during the season—and 5–10 trainings hours of recreational time, including continuous endurance training such as cycling and running, functional strength training and individual training to improve muscle disbalances. No adverse cardiac events or arrhythmias were reported in any individual athlete’s history.

### 2.3. Measurements

As part of the sports medicine evaluation in the accompanying performance center, we performed twelve-lead electrocardiograms (ECGs) in a lying position with 50 mm/s (CardioSoft V6.73, GE Medical Systems, Munich, Germany) to define resting heart rate in beats per minute (bpm) after 5 min. All participants were additionally evaluated for BMI (kg/m^2^), and resting blood pressure level (in mmHg) after resting for five minutes in a supine position. 

The CPET was conducted in accordance with the recommendations of the American Heart Association (AHA) [25], either on a bicycle or on a treadmill, as predetermined by the national winter sport discipline association. The CPET step-wise protocol started with a workload of 80 Watts and the workload was increased by 40 Watts every 3 min until volitional exhaustion. Alternatively, the treadmill tests started with a workload of 8 km/h for 3 min and then increasing the speed by 1 km/h every 3 min. We continuously recorded all performance data from the CPET measurements and twelve-lead ECG and blood pressure. To define peak performance criteria for CPET analysis, several factors were taken into consideration. First, each participant was analyzed to determine their peri- and post-exercise lactate level with a capillary blood analysis from the earlobe at each workload step, individual anaerobic threshold (4 mmol/L), and their recovery time (maximum 15 mmol/L). Furthermore, levelling off of the VO_2 max_, reaching 85% of the individual maximum predicted heart rate (220 bpm minus age in years), a respiratory exchange rate (RER) of 1.15 at peak performance, and individual assumed exercise time of CPET duration were the factors focused on in the CPET analyses. The specific athlete’s exertion level was analyzed according to the Borg RPE scale (Values ≥ 17). To define an individual athlete’s maximal CPET effort in our elite winter sport athletes, a minimum of three of the above mentioned criteria were taken into consideration [15]. The CPET data were analyzed and the peak oxygen pulse (Oxygen pulse _maximum_) as well as the oxygen pulse at VT2 (Oxygen pulse_VT2_) were calculated by dividing the derived VO_2_ by the maximum heart rate or the heart rate at VT2 during exercise [26].

Additional information about the management of training schedules and sessions was evaluated by interviewing the athletes and the training staff. In particular, detailed information about the individual athletes’ training schedule such as endurance training (ET), strength training (ST), movement training skills and recreational training conditioning programs was evaluated in order to understand the differences between our participating three cohorts of world elite winter sports athletes.

### 2.4. Statistical Analysis

Our data were analyzed with Graph Pad Prism 8.2.1(279) (Graph Pad Software; San Diego, CA, USA). All data were assessed for normal distribution by analyzing the data by means of Kolmogorov–Smirnov normality testing. Afterwards, we evaluated our numerical data group comparisons for nine Ski-Mo athletes (5 male, 4 women), ten NCC athletes (6 male, 4 women), and twelve elite Bia athletes (6 male, 6 women) using Mann–Whitney testing. Afterwards, a gender-specific analysis for the interesting parameters was utilized equally. Statistical significance was accepted as *p* ≤ 0.05.

### 2.5. Ethical Consideration

The local ethics committee of the University of Nurnberg-Erlangen approved the study protocol (17_21 B). The study was conducted in conformity with the declaration of Helsinki and Good Clinical Practice [27]. Before any trial-related activities, our participating athletes gave their written informed consent and were informed about the study protocol as well as the following measurements.

## 3. Results

A total of thirty-one young professional winter sports athletes were examined. The matched data of the three different participating winter sports were compared for different anthropometric data and CPET performance parameters. The recorded athletes’ heart rate at baseline (bpm) as well as the resting blood pressure level (mmHg) were quite low in this elite winter sport population. The athlete’s individual heart rate response at VT2 as well as at maximum effort during CPET are represented in Table 1. 

All athletes showed, as expected in elite winter sport athletes, excellent performance data in the CPET (Table 2). Updated Olympic-medal-level performance benchmark data served as a reference [28].

In this context, the Ski-Mo athletes showed a significantly higher respiratory minute volume (VE _VT2_) at the second ventilatory threshold (VT2) compared to Bia and NCC athletes, but in the end the highest maximum respiratory minute volume (VE _maximum_) was identified for the Bia athletes, who showed significant differences in comparison the Ski-Mo athletes.

No significant differences could be revealed for the maximum ventilatory oxygen uptake (VO_2 maximum_) in our three different elite winter sport disciplines. The ventilatory oxygen uptake (VO_2_) was calculated as an index parameter at the VT2 (VO_2_/kg _VT2_) as well as at the maximum performance level (VO_2_/kg _maximum_). Ski-Mo athletes showed significantly higher performance values for the VO_2_/kg _VT2_ in comparison to the NCC and Bia athletes (Figure 1), whereby the performance data analyses of the three participating winter sport professionals did not reach significant differences regarding the VO_2_/kg _maximum_ benchmark.

The Oxygen pulse _maximum_ as well as the Oxygen pulse _VT2_ were analyzed and the results are presented in detail in Table 2. Focusing on this parameter, NCC and Bia athletes showed significantly higher Oxygen pulse _maximum_ values compared to Ski-Mo athletes (Figure 2). 

In this context, analyzing the sex-specific differences of these parameters across the three different disciplines, the female Ski-Mo athletes showed the highest respiratory minute volume (VE) at the second ventilatory threshold (VT2) (*p* = 0.0286), representing significantly higher performance values for the VO_2_/kg _VT2_ in comparison to the female NCC and Bia athletes (*p* = 0.0286 & 0.0381, results shown in Table 2). The peak oxygen pulse (Oxygen pulse maximum) was highest for female Bia athletes compared to female Ski-Mo and NCC athletes (*p* = 0.0190).

The male athletes did not differ significantly with regard to the highest respiratory minute volume (VE) at the second ventilatory threshold (VT2), whereby male Ski-Mo athletes showed the best performance values in comparison to the NCC and Bia athletes (*p* = 0.0519 & 0.0823), but in the end the highest maximum respiratory minute volume (VE _maximum_) was identified for the male Bia athletes, showing significant differences in comparison to male Ski-Mo athletes (*p* = 0.0087, results shown in Table 2). Additionally, maximum ventilatory oxygen uptake (VO_2 maximum_) was significantly higher for male Bia athletes compared to male Ski-Mo athletes (*p* = 0.0087, Figure 3), although no significant differences were revealed for NCC compared to Bia athletes. The significantly best performance data referring the peak oxygen pulse (Oxygen pulse _maximum_) in the three participating male athletes could be found for the male NCC and Bia athletes (*p* = 0.0303 & 0.0260, results shown in Table 2).

## 4. Discussion

The present descriptive preliminary study reported the sport-specific physiological performance with laboratory CPET analyses as predictors of performance in world elite professional winter sports athletes. The analyzed winter sports in this study have been reported as the most endurant competitive sports in several studies before, as their intense energy demands, the environmental circumstances as well as their trainings schedule—including endurance training (ET) and strength training (ST) components—are very challenging for competitive athletes in general [3,4,5,9,10,29]. Various parameters influencing performance in elite winter sport athletes are reported in several studies and are used for predictors of performance in these athletes, especially with an exceptionally high aerobic energy turnover, an excellent anaerobic power, and different strength and speed qualities [8,20,23,30,31,32,33]. The repeated intensity fluctuations and their physiological adaptions are possible due to the nature of cross-country ski competitions, which involve recovery downhill episodes, high intensity sprints, and uphill racing [29]. The participants in this study had slightly lower CPET values, defined as VO_2_/kg _maximum_, compared to Olympic-medal-level benchmarks from 1990–2013 [28]. These results emphasize the quality of the highly endurant individual athlete’s CPET performance in our descriptive report.

In this context, in our descriptive preliminary report, the Ski-Mo athletes had a significantly higher respiratory minute volume at the VT2 (VE _VT2_) compared to Bia and NCC athletes. This finding is a known phenomenon, because Ski-Mo athletes are famous for their pronounced aerobic capacity required by repeated intensity fluctuations, which is an important requirement for their success, as demonstrated by positive correlations of race performance and oxygen uptake levels and heart rate variability at VT2 in previous research [12,34]. Even previously reported physiological demands in NCC athletes require high maximal oxygen uptakes, as well as high anaerobic thresholds, although anaerobic thresholds play a less important role, confirming the data of our participating athletes [8]. Moreover, due to the considerable mechanical work of demanding uphill locomotion during competition, an inverse correlation between the weight of the athlete and race performance has been demonstrated [12]. These findings are supported by the results of our descriptive reporting, where the Ski-Mo athletes represent the homogenous youngest and physically smallest group of male and female athletes with less lifetime training hours compared to the career of NCC and Bia athletes [7]. Our findings support the thesis of training inducing physiological athlete’s adaption due to various training stimuli and training specificity for optimal performance [8], nevertheless our results have to be handled with care. 

Additionally interesting aspects, which have to be mentioned, are reported by Schupfner et al., who found a positive correlation of the increasing age of German Nordic combined athletes with their performance in the World cup, the maximum oxygen uptake and their individual anaerobic threshold [21]. In this research, a higher BMI proved advantageous in terms of an individual athlete’s performance in the World cup [21]. These findings are supported by the results of our descriptive reporting, whereby NCC and Bia athletes represent the more weighty, taller and more experienced athletes and in the end showed higher physiological performance parameters in the laboratory exercise testing, defined as VE _maximum_, VO_2 maximum_, Oxygen pulse _VT2_, and Oxygen pulse _maximum_. Our results are even supported by Rusko et al., who revealed an increase in the VO_2 maximum_ in NCC athletes with age, and lifetime training hours accumulation, enhanced volume of training, and volume of intensive training [35]. 

Additionally, a significantly higher maximum ventilatory oxygen uptake (VO_2 maximum_) and higher peak oxygen pulse (Oxygen pulse _maximum_) were revealed for the Bia and NCC athletes in comparison to Ski-Mo athletes across our three different elite winter sport disciplines. Therefore, in our opinion, several factors might play an important role. On the one hand, as elucidated in our previous research on cardiac remodeling in elite winter sport athletes, Ski Mo athletes, as the physically smallest athletes, showed different morphological and functional cardiac remodeling compared to the NCC and Bia athletes [7]. Therefore, lower left ventricular (LV) Mass index, less left atrial remodeling as measured by the left atrial (LA) volume index, and lower values for the LV global longitudinal strain (GLS) were proven for the Ski-Mo athletes. Next to these cardiac remodeling influencing factors, the Ski-Mo athletes are known for their excellent aerobic capacity [11,12,34,36], but represent the youngest and physically smallest athletes with a major ET component in their training schedule. On the basis of this excellent respiratory capacity, the Ski-Mo athletes performed very well at the VT2, but the NCC and Bia athletes represent the physically stronger and more experienced participants to create higher maximum ventilatory oxygen uptake. Derived by the echocardiographic remodeling, the higher individual maximum ventilatory uptake and the comparable heart rate response at maximum effort, the Bia and NCC athletes seem to be able to create a higher individual maximum oxygen pulse. 

All participating elite athletes are professional athletes with a total amount of 20–25 training hours per week. As reported before, the main training focus in Ski-Mo athletes is on endurance training sessions, whereas strength-training sessions represent an essential part in NCC and Bia athlete’s training schedule resulting in an individualized sport-specific cardiac remodeling [7]. In this context, previous research on this topic reported no significant changes in elite NCC athletes for the VO_2 peak_ measurements across the season with variable training schedule elements, but revealed positive effects for performance and oxygen-cost and oxygen-deficit in these athletes [31]. Analyzing different training strategies of elite NCC and Nordic-combined athletes, previous studies emphasized the importance of training with low intensity with a moderate volume of ST and speed training for athlete’s performance [37] as well as high-intensity ST and ET [38]. These varying circumstances have to be taken into consideration while investigating an individual athlete’s performance in CPET, which represents the preferred tool in sports medicine for quantitative and qualitative assessment of the metabolic and cardiopulmonary response to exercise in athletes.

Summarizing the obtained results from our descriptive reporting study, we were able to provide new evidence for significant differences in CPET values in participating winter sport athletes. These differences are defined on the one hand by the excellent respiratory capacity at the VT2, especially in Ski-Mo athletes, as well as the previously shown different morphological and functional cardiac remodeling in elite winter sport athletes [7]. The oxygen pulse, calculated by dividing the derived VO_2_ by heart rate, is considered to correlate with the athlete’s stroke volume. Regarding this cardiac parameter, the sport-specific LA and LV remodeling in NCC and Bia athletes seems to determine their enhanced performance at VT2 and maximum effort in comparison to Ski-Mo athletes. Next to the influencing parameters, such as athlete’s anthropometric, muscular and training schedule variables, these obtained findings from CPET data might influence the individual sport-specific training planning in these athletes in the future. Based on our investigative results, it might be recommended for NCC and Bia athletes to improve their respiratory capacity at the VT2 by specific training sessions, whereas our physically small and young Ski-Mo athletes might have to focus on trainings sessions to improve their peak cardiac performance capacity along their lifetime training career.

Our descriptive reporting study has several limitations. First of all, the number of participating winter sport professionals is relatively small which is due to the fact that we only evaluated high level athletes from the German national teams competing at world-class events. Secondly, in our participating cohort, Ski-Mo athletes represented the youngest and physically smallest athlete category, implying less lifetime training hours and competition frequencies. Furthermore, the CPET analyses of the participating athletes were performed in a multicenter study design in the preseason preparation time, so that an interobserver variability with respect to data acquirement must be considered. Although laboratory-based measurements can effectively assess the individuals athlete’s performance, conclusions drawn from these results might not be fully applicable to the on-snow performance of these athletes, as mentioned in previous scientific research [23]. Various studies have investigated the ability of laboratory- and field-based tests to predict elite winter sport athletes’ performance, suggesting that both field-based and laboratory performance indices provide valid predictions of winter sport athletes’ training and race performance [5,24,33,36]. 

## 5. Conclusions

This report provides new evidence that in different world elite winter sport professionals, significant differences in CPET parameters can be demonstrated, against the background of an athlete’s physique and anthropometric data as well as training control and frequency. 

Our results might serve as a preliminary report and will have to be handled with care due to the mentioned limitations. Therefore, our results—in general as well as in the gender-specific subgroup analyses—can reveal sport-specific individual differences in the CPET performance of our three participating world elite winter sport professionals, especially due to the VE _VT2_, VE _maximum_, VO_2_/kg _VT2_, Oxygen pulse _VT2_, and Oxygen pulse _maximum_. These differences between the groups might be defined by an athlete’s individual respiratory capacity as well as the functional and structural sport specific cardiac remodeling. Nevertheless, when interpreting an athlete’s individual performance in CPET, the impact of athlete’s physique, the sport-specific cardiac remodeling, the main emphasis in an athlete’s training schedule, and accumulation of lifetime training hours have to be taken into consideration.

Future studies with a greater number of participating winter sport professionals and long term follow up periods have to be considered to elucidate these findings and strengthen the scientific basis of evidence.

## Figures and Tables

**Figure 1 ijerph-19-05620-f001:**
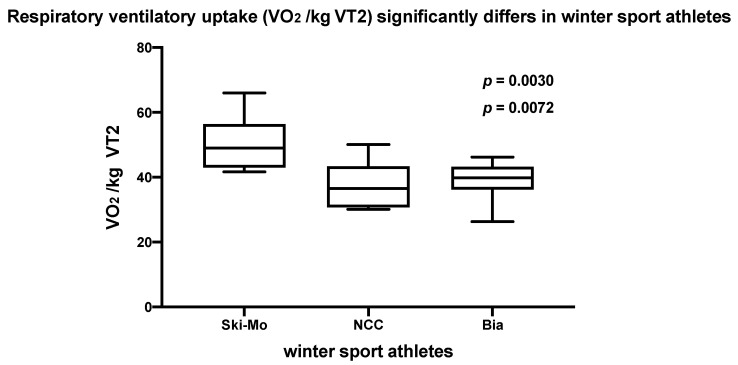
Respiratory ventilatory uptake (VO_2_/kg _VT2_) significantly differs in elite winter sports athletes (Ski-Mo vs. NCC, *p* = 0.0072; Ski-Mo vs. Bia, *p* = 0.0030).

**Figure 2 ijerph-19-05620-f002:**
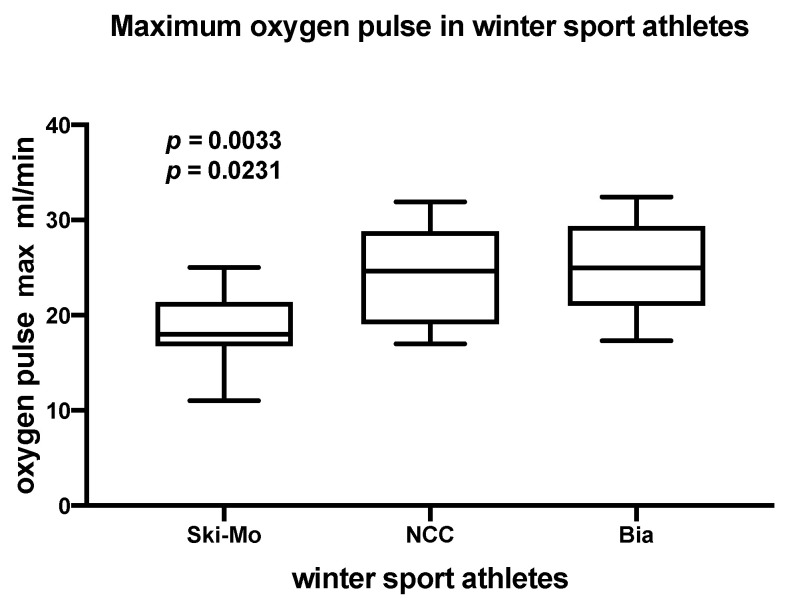
Peak oxygen pulse (Oxygen pulse _maximum_) significantly differs in elite winter sports athletes (Ski-Mo vs. NCC, *p* = 0.0231; Ski-Mo vs. Bia, *p* = 0.0033).

**Figure 3 ijerph-19-05620-f003:**
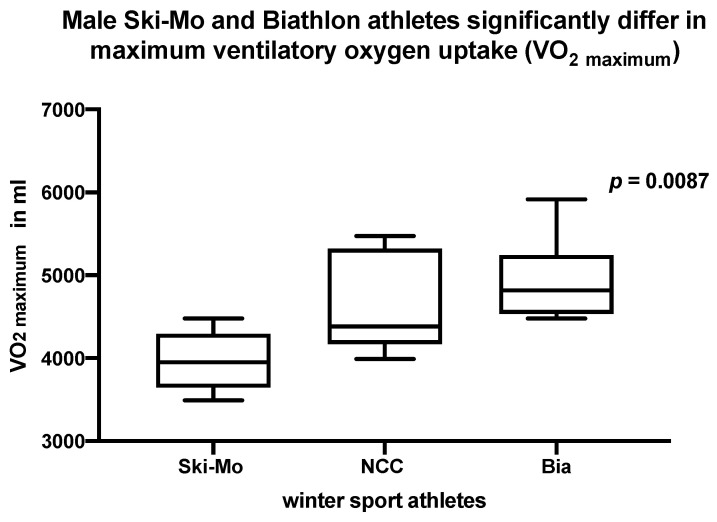
Maximum ventilatory oxygen uptake (VO_2 maximum_) significantly differs in elite male Ski-Mo and Bia athletes (*p* = 0.0087).

**Table 1 ijerph-19-05620-t001:** Anthropometric data of the elite winter sport athletes.

	Ski-Mo n = 9		NCC n = 10		Biathletes n = 12	
	Male n = 5	Female n = 4	Male n = 6	Female n = 4	Male n = 6	Female n = 6
Age (y)	21.4 ± 1.8	20.8 ± 2.4	26.3 ± 4.1	25.5 ± 0.5	27.3 ± 3.6	29.0 ± 3.2
Height (cm)	178.0 ± 3.9	163.5 ± 8.8	181.3 ± 4.7	171.2 ± 5.8	180.9 ± 5.1	172.8 ± 3.7
Weight (kg)	66.5 ± 0.8	53.2 ± 6.5	72.0 ± 3.0	63.4 ± 5.9	77.1 ± 3.7	62.5 ± 4.1
BMI (kg/m^2^)	19.9 ± 1.4	19.8 ± 0.4	22.0 ± 1.1	21.6 ± 1.2	23.6 ± 0.9	20.9 ± 1.0
resting blood pressure systolic/diastolic (mmHg)	118 ± 5.4 78 ± 4.0	100 ± 8.2 72 ± 1.5	125 ± 8.3 78 ± 2.9	105 ± 7.2 71 ± 3.8	117 ± 7.6 77 ± 2.2	108 ± 6.2 70 ± 3.3
resting heart rate (bpm)	41 ± 4.6	44 ± 4.5	42 ± 3.6	46 ± 5.1	41 ± 4.2	45 ± 5.1
heart rate VT2 (bpm)	133 ± 2.2	132.3 ± 1.9	136.3 ± 11.6	128.3 ± 9.1	148.5 ± 20.9	134.5 ± 8.1
maximum heart rate (bpm)	185.6 ± 6.3	171.8 ± 2.5	183 ± 14.3	173.8 ± 4.0	179.5 ± 10.3	181.0 ± 12.9
BSA (body surface area m^2^)	1.70 ± 0.06	1.61 ± 0.12	1.88 ± 0.04	1.81 ±0.07	1.92 ± 0.04	1.77 ± 0.05

Data are presented as mean with standard deviation.

**Table 2 ijerph-19-05620-t002:** Cardiopulmonary exercise testing (CPET) data of elite winter sport athletes.

	Ski-Mo (I)		NCC (II)		Biathletes (III)		*p*-Value Male	*p*-Value Female	Overall *p*-Value
	Male	Female	Male	Female	Male	Female			
**VE _VT2_ (L)**	96.4 ± 15.5	77.0 ± 11.2	71.3 ± 15.9	55.1 ± 6.9	80.2 ± 7.2	62.7 ± 13.0	**ns**	Ski-Mo vs. NCC **0.0286 ***	Ski-Mo vs. NCC **0.0076 ***
	**87.8 ± 16.5**		**64.8 ± 15.0**		**71.4 ± 13.6**				Ski-Mo vs. Bia **0.0339 ***
**VE _maximum_ (L)**	134.9 ± 24.6	109.2 ± 20.6	166.2 ± 28.4	118.2± 23.8	175.8 ± 11.7	125.4 ± 9.1	Ski-Mo vs. Bia	**ns**	Ski-Mo vs. Bia
	**123.5 ± 25.4**		**147.0 ± 35.4**		**150.6 ± 28.1**		**0.0087 ***		**0.0409 ***
**VO_2 maximum_ (mL)**	3964.8 ± 367.8	3021.3 ± 515.1	4620.8 ± 603.8	3315.3 ± 576.0	4935.2 ± 525.1	3555.7± 274.7	Ski-Mo vs. Bia **0.0087 ***	Ski-Mo vs. Bia **0.0381 ***	**ns**
	**3545.4 ± 643.7**		**4098.6 ± 876.2**		**4245.4 ± 823.8**				
**VO_2_/kg _VT2_ (mL/kg)**	52.3 ± 9.7	47.5 ± 5.8	41.4 ± 7.3	33.0 ± 2.8	41.5 ± 4.5	36.6 ± 6.1	**ns**	Ski-Mo vs. NCC **0.0286 ***	Ski-Mo vs. NCC **0.0072 ***
	**50.2 ± 8.1**		**38.1 ± 7.1**		**39.1 ± 5.6**			Ski-Mo vs. Bia **0.0381 ***	Ski-Mo vs. Bia **0.0030 ***
**VO_2_/kg _maximum_ (mL/kg)**	65.0 ± 7.9	57.4 ± 4.5	64.5 ± 7.1	52.7 ± 4.9	64.6 ± 4.4	57.4 ± 2.3	**ns**	**ns**	**ns**
	**61.6 ± 7.5**		**59.7 ± 8.6**		**61.0 ± 5.0**				
**Oxygen pulse _VT2_ (mL/min)**	18.4 ± 2.6	14.4 ± 2.3	21.8 ± 3.5	16.3 ± 2.4	21.6 ± 2.6	16.9 ± 2.7	**ns**	**ns**	Ski-Mo vs. Bia **0.0426 ***
	**16.6 ± 3.1**		**19.6 ± 4.1**		**19.2 ± 3.5**				
**Oxygen pulse _maximum_ (mL/min)**	20.8 ± 3.0	15.6 ± 3.1	26.9 ± 4.2	19.4 ± 3.1	27.8 ± 3.2	22.8± 5.4	Ski-Mo vs. NCC **0.0303 *** Ski-Mo vs. Bia **0.0260 ***	Ski-Mo vs. Bia **0.0190 ***	Ski-Mo vs. NCC **0.0231 *** Ski-Mo vs. Bia **0.0033 ***
	**18.5 ± 4.0**		**23.9 ± 5.3**		**25.3 ± 4.9**				

Data are presented as mean with standard deviation. *p* value *****, statistically significant (*p* < 0.05). Abbreviations: CPET, cardiopulmonary exercise testing; Ski-Mo, Ski-mountaineering; NCC, Nordic Cross-Country; VE, respiratory minute volume; VT2, second ventilatory threshold; VO_2_, ventilatory oxygen uptake; VO_2_/kg, ventilatory oxygen uptake per kilogram; L, liter; mL, milli-liter; min, minute; ns, not significant.

## Data Availability

Individual anonymized data supporting the analyses of this study contained in this manuscript will be made available upon reasonable written request from researchers whose proposed use of data for a specific purpose has been approved.

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
