# Peer review of "Physiological Aspects of World Elite Competitive German Winter Sport Athletes"

_ijerph, 2022, doi:10.3390/ijerph19095620_

Round 1

Reviewer 1 Report

The main aim of the descriptive preliminary report „Physiological aspects of world elite competitive German winter sport athletes“ is to elucidate the individual physiological remodeling by CPET data analyses in German elite competitive winter sports athletes.

The study is interesting. I would like to appretiate the efforts of the authors. However, some facts need to be explained.

Major comment:

I recommend better organization of the text, I recommend to divide the chapter Materials and Methods into: Study design, Participants, Procedure (or Measurements), Statistical Analysis, … In the participants section I recommend better characterization of participants of the study.

The discussion describes the differences between the groups and I recommend that you better formulate the conclusion and include what the differences between the groups are and how you explain these differences. This is so that there is some output to the practical application. In conclusion, should it be highlighted what is the practical significance of the results found?

All participants in the study were at comparable performance level in their sports?

Minor comments:

Line 17–21: 12+10+10=31 ??? sportsmen

Line 71: 1 Materials and methods, correctly - section number 2.

Under Table 1 is extra description Table 1.

Is it necessary to use the image name above the image (Figures 1, 2, 3) when it is described below the image?

I recommend explaining in the text why the results of the NCC group are not presented in Figure 2 and Figure 3.

Author Response

Dear Editor,

Dear Reviewers,

Thank you very much for reviewing our manuscript entitled “Physiological aspects of world elite competitive German winter sport athletes”. Please find below a point-to-point response to the specific comments.

Reviewer 1

Comments and Suggestions for Authors

The main aim of the descriptive preliminary report „Physiological aspects of world elite competitive German winter sport athletes“ is to elucidate the individual physiological remodeling by CPET data analyses in German elite competitive winter sports athletes.

The study is interesting. I would like to appreciate the efforts of the authors. However, some facts need to be explained.

Dear Reviewer,

we highly appreciate the willingness to review our manuscript and also express our thanks for the comments and positive feedback for the authors. Please find below a point-to-point response to the specific comments.

Major comment:

I recommend better organization of the text, I recommend to divide the chapter Materials and Methods into: Study design, Participants, Procedure (or Measurements), Statistical Analysis, … In the participants section I recommend better characterization of participants of the study.

Thank you very much for the comment on the organization of the Chapter Material and Methods. We appreciate the reviewer´s suggestions to improve the structure and quality of our work. In the revised version of our manuscript these major comments were processed and our manuscript was amended as following on page 2, line 105 to page 3, line 202:

“2. Materials and Methods

Study design

This was a multicenter retrospective analysis of CPET data in 31 elite winter sport athletes, which were obtained in 2021 during the annual medical sports medicine check-up. The matched data of the elite winter sports athletes (14 women, 17 male athletes, age: 18-32 years) were compared for different CPET parameters, athlete´s anthropometric data and sport specific training aspects.

Participants

As study population, thirty-one young elite winter sport professionals, all active members of the German National winter sport Team, participating in world championships as well as World cups, were examined during the season of 2021. All participants (14 women, 17 male athletes, age: 18-32 years) in this study were at comparable performance level in their sport discipline. All participants underwent an individual sports medicine check-up in their sports medicine performance center - Institute for Applied Exercise Science, University Leipzig or Interdisciplinary Center of Sportsmedicine, Klinikum Bamberg in the preseason preparation. No participant was infected with Corona virus during the severe COVID-19 pandemic situation and so none had to be excluded from the annual performance testing. Male athletes had an age of 18-32 years, had a height between 175-186 cm, were of weight between 65-81 kg and had a Body Mass Index (BMI) of 18-25 kg/m2. The participating female athletes were between 18-31 years, had a height of 154-176 cm, had a weight between 47-69 kg and showed a BMI of 18-23 kg/m2. All participants were professional athletes with a total amount of 20-25 training hours per week - based on the individual high volume training schedule during the season - and an amount of 5-10 trainings hours of recreational time, including continuous endurance training as cycling and running, functional strength training and individual training to improve muscle dysbalances. No adverse cardiac events or arrhythmias have been reported in the individual athlete´s history.

Measurements

As part of the sports medicine evaluation in the accompanying performance center, we performed twelve-lead electrocardiograms (ECGs) in lying position with 50 mm/s (CardioSoft V6.73, GE Medical Systems, Germany) to define resting heart rate in beats per minute (bpm) after 5 minutes. All participants were additionally evaluated for BMI (kg/m2), and resting blood pressure level (in mmHg) after resting for five minutes in a supine position.

The CPET was conducted in accordance to the recommendations of the American Heart Association (AHA) [25], either on the bicycle or on the treadmill, as predetermined by the national winter sport discipline association. The CPET step-wise protocol started with a workload of 80 Watts and the workload was increased by 40 Watts every 3 minutes until volitional exhaustion. Alternatively, the treadmill tests started with a workload of 8 km/h for 3 min increasing speed by 1 km/h every 3 minutes. We continuously recorded all performance data from the CPET measurements, additionally twelve-lead ECG and blood pressure. To define peak performance criteria for CPET analysis several factors were taken into consideration. First each participant was analyzed for peri- and post-exercise lactate level by capillary blood analysis from the earlobe at each workload step, individual anaerobic threshold (4 mmol/l), and in the recovery time (maximum 15 mmol/l). Furthermore the VO2 max levelling off in the CPET analyses, reaching 85% of the individual maximum predicted heart rate (220 bpm minus age in years), reaching a respiratory exchange rate (RER) 1.15 at peak performance, and reaching individual assumed exercise time of CPET duration were focused in CPET analyses. The specific athlete´s exertion level was analyzed analogue to the Borg RPE scale (Values ≥ 17). To define an individual satisfying athlete´s maximal CPET effort in our elite winter sport athletes a minimum of three of the above mentioned criteria were taken into consideration [15]. The CPET data were analyzed and the peak oxygen pulse (Oxygen pulse maximum) as well as the oxygen pulse at VT2 (Oxygen pulse VT2) were calculated by dividing derived VO2 by the maximum heart rate or the heart rate at VT2 during exercise [26].

Additional information about the management of training schedules and sessions was evaluated by interviewing the athletes and the training staff. In particular, detailed information about the individual athletes´ training schedule as endurance training (ET), strength training (ST), movement training skills and recreational training conditioning programs was evaluated in order to understand the differences between our participating three cohorts of world elite winter sports athletes.

Statistical analysis

Our data were analyzed with Graph Pad Prism 8.2.1(279) (Graph Pad Software; USA). All data were assessed for normal distribution by analyzing the data by means of Kolmogorov-Smirnov normality testing. Afterwards we evaluated our numerical data group comparisons for nine Ski-Mo athletes (5 male, 4 women), ten NCC athletes (6 male, 4 women), and twelve elite Bia athletes (6 male, 6 women) using the analysis Mann Whitney testing. Afterwards, a gender specific analysis for the interestingly parameters was utilized equally. Statistical significance was accepted by p ≤ 0.05.

Ethical consideration

The local ethics committee of the University of Nurnberg-Erlangen approved the study protocol (17_21 B). The study was conducted in conformity with the declaration of Helsinki and Good Clinical Practice [27]. Before any trial related activities, our participating athletes gave their written informed consent and were informed about the study protocol as well as following measurements.”

The discussion describes the differences between the groups and I recommend that you better formulate the conclusion and include what the differences between the groups are and how you explain these differences. This is so that there is some output to the practical application. In conclusion, should it be highlighted what is the practical significance of the results found?

We want to thank the reviewer for addressing this important point to us. The reviewer is absolutely right that the described differences between the groups should be discussed more widely and the potential output to practical application should be mentioned in the discussion. Thank you very much.

Therefore, our revised manuscript was amended as following on page 9, line 740 to 759:

“These varying circumstances have to be taken into consideration while investigating the individual athlete´s performance in CPET, which represents the preferred tool in sports medicine for quantitative and qualitative assessment of the metabolic and cardiopulmonary response to exercise in athletes.

Summarizing the obtained results from our descriptive reporting study, we were able to provide new evidence for significant differences in CPET values in participating winter sport athletes. These differences are defined on the one hand by the excellent respiratory capacity at the VT2, especially in Ski-Mo athletes, as well as the previously shown different morphological and functional cardiac remodeling in elite winter sport athletes. The oxygen pulse, calculated by dividing derived VO2 by the heart rate, is considered to correlate with the athlete´s stroke volume. Regarding this cardiac parameter, the sport specific LA and LV remodeling in NCC and Bia athletes seem to determine their enhanced performance at VT2 and maximum effort in comparison to Ski-Mo athletes. Next to influencing parameters, such as athlete´s anthropometric, muscular and training schedule variables, these obtained findings from CPET data might influence the individual sport specific training planning in these athletes in the future. Based on our investigative results, it might be recommended for NCC and Bia athletes to improve their respiratory capacity at the VT2 by specific training sessions, whereas our physically small and young Ski-Mo athletes might have to focus on trainings sessions to improve their peak cardiac performance capacity along life time training career.”

Therefore, our revised manuscript was amended as following on page 9, line 776 to page 10, line 823:

“5. Conclusions

This report provides new evidence that in different world elite winter sport professionals significant differences in CPET parameters can be demonstrated, against the background of athlete´s physique and anthropometric data as well as training control and frequency.

Our results might serve as a preliminary report and have to be handled with care due to the mentioned limitations. Therefore our results – in general as well as in the gender specific subgroup analyses – can reveal sport specific individual differences in the CPET performance of our three participating world elite winter sport professionals, especially due to the VE VT2, VE maximum, VO2/kg VT2, Oxygen pulse VT2, and Oxygen pulse maximum. These differences between the groups might be defined by the athlete´s individual respiratory capacity as well as the functional and structural sport specific cardiac remodeling. Nevertheless interpreting the athlete´s individual performance in CPET the impact of athlete´s physique, the sport specific cardiac remodeling, the different main emphasis in training schedule, and accumulation of life time training hours have to be taken into consideration.

Future studies with a greater number of participating winter sport professionals and long term follow up periods have to be considered to elucidate these findings and strengthen the scientific evidence base.”

All participants in the study were at comparable performance level in their sports?

We want to thank the reviewer for this important comment. As the participating athletes were at comparable level in their sports, this fact was added in the Material and Methods section on page 2, line 116 to 118:

“All participants (14 women, 17 male athletes, age: 18-32 years) in this study were at comparable performance level in their sport discipline."

Minor comments:

Line 17–21: 12+10+10=31 ??? sportsmen

Thank you very much for the comment on our inaccuracy, we corrected the number of athletes in the revised manuscript on page 1, line 17 to 21:

“Nine Ski mountaineering (Ski-Mo), ten Nordic-Cross Country (NCC) and twelve world elite biathlon (Bia) athletes were evaluated for cardiopulmonary exercise test (CPET) performance as the primary aim of our descriptive preliminary report. A multicenter retrospective analysis of CPET data was performed in 31 elite winter sports athletes, which were obtained in 2021 during the annual medical examination.“

Line 71: 1 Materials and methods, correctly - section number 2.

Thank you very much for the comment on our inaccuracy, we corrected the section number:

  1. Material and Methods

Under Table 1 is extra description Table 1.

We want to thank the reviewer for this comment on our inaccuracy. In our revised manuscript we moved the title up to the heading and amended our manuscript as following on page 4, Table 1 and page 5, Table 2. Furthermore the description below the images is deleted in the revised version.

Is it necessary to use the image name above the image (Figures 1, 2, 3) when it is described below the image?

We want to thank the reviewer for this comment on our inaccuracy. In our revised manuscript several changes were performed in the Figures 1, 2, 3 – the figure captions were modified and p-values were added in order to picture the results more clearly. We improved our revised manuscript referring these inaccuracies and are sorry for the inconveniences.

I recommend explaining in the text why the results of the NCC group are not presented in Figure 2 and Figure 3.

We want to thank the reviewer for this comment on our inaccuracy and we apologize for the inconveniencies.

Figure 2 was revised and the detailed data of all three participating winter sport athletes were added in the current Figure 2 and the manuscript was amended as following on page 5, line 425 to 429:

“The Oxygen pulse maximum as well as the Oxygen pulse VT2 were analyzed and the results are presented in detail in Table 2. Focusing this parameter, NCC and Bia athletes showed significant higher Oxygen pulse maximum values compared to Ski-Mo athletes (Figure 2).“

Figure 3 was revised and the detailed data are shown in the current figure. Additionally our results section was amended as following to improve the quality of our manuscript.

Page 7, line 480 to 486:

“Additionally, maximum ventilatory oxygen uptake (VO2 maximum) was significantly higher for male Bia athletes compared to male Ski- Mo (p = 0.0087, Figure 3), no significant differences were revealed for NCC compared to Bia athletes. The significant best performance data referring the peak oxygen pulse (Oxygen pulse maximum) in the three participating male athletes could be proven for the male NCC and Bia athletes (p = 0.0303 & 0.0260, results shown in Table 2).”

Reviewer 2 Report

See attached document for suggestions changes/improvements. 

Author Response

Reviewer 2

Dear Editor,

Dear Reviewer,

Thank you very much for reviewing our manuscript entitled “Physiological aspects of world elite competitive German winter sport athletes”. Please find below a point-to-point response to the specific comments.

Manuscript Review – Physiological aspects of world elite competitive German winter sport athletes.

Thank you for your contributions to the field.

Many of the corrections are grammar fixes related to effectively communicating in English. Therefore, I provided some suggestions for corrections. I also suggest providing clarity and consistency in presenting results (e.g., data in tables and figures). I hope my comments below make sense and add to the clarity of your manuscript.

We would like to thank the reviewer for taking the time to read our manuscript and we appreciate the suggestions addressing important points to improve the quality of our work.

We would like to thank the reviewer for addressing this important point to review the grammar as well as English formatting and thank for the suggestions from the reviewer. We are aware about this inaccuracy and revised our manuscript by an English native.

We appreciate all the suggestions from the reviewer. Please find below a point-to-point response to the specific comments.

Overall

  1. You use the phrase “remodeling by CPET analyses”, in the abstract and conclusions, although this doesn’t make sense to me as it is written. CPET is a test and can indicate fitness level and an overall improvement in the functioning of the body but does not give indication as to the specific parts of the body that have improved/changed. These sections of the manuscript probably just need to be reworded/edited so it is clear.

Thank you very much for the comment. The reviewer is absolutely right that the phrase has to be removed and fact has to be stated clearly in our revised manuscript.

Therefore the abstract was amended as following, page  1, line 28 to 30:

“This report provides new evidence that in different world elite winter sport professionals significant differences in CPET parameters can be demonstrated, against the background of athlete´s physique as well as training control and frequency.“

and the conclusion on page 9, line 776 to 779:

“This report provides new evidence that in different world elite winter sport professionals significant differences in CPET parameters can be demonstrated, against the background of athlete´s physique and anthropometric data as well as training control and frequency.”

2. You use the word “endurant” in several parts of the manuscript. Did you intend to use this word, or did you intend to use the word “endurance”? The definition of endurant is “capable of enduring adversity, severity, or hardship” so if that is what you meant then it makes sense to use this word.

We appreciate the interesting question from the reviewer. Indeed, in our opinion we intended to use the word “endurant” to highlight the very demanding character of the analyzed winter sports and to pay attention to the extraordinary character of the individual athlete´s performance.

Abstract

  1. Line 17: The order of BIA, NCC, and Ski-Mo is different here than in other parts of the manuscript (e.g., Line 35, Lines 120-121, Table 1). Use a consistent order in all parts of the manuscript.

We want to thank the reviewer for this important comment. The reviewer is absolutely right that it should be performed a consistent order of the different winter sport disciplines in all parts of the manuscript. Therefore, our revised manuscript was amended as following,

on page 1, line 17 to 18:

“Nine Ski mountaineering (Ski-Mo), ten Nordic-Cross Country (NCC) and twelve world elite biathlon (Bia) athletes…”

On page 1, line 35:

“Ski mountaineering (Ski-Mo), Nordic-Cross country (NCC) and Biathlon (Bia) are…”

Furthermore in the revised manuscript a consistent order of the participating athletes was pictured in Table 1 and Table 2.

2. Line 18: Add the word “test”. “...cardiopulmonary exercise test (CPET) performance...”

We want to thank the reviewer for making us aware about this missing word. We amended our revised manuscript on page 1, line 17 to 19:

“Nine Ski mountaineering (Ski-Mo), ten Nordic-Cross Country (NCC) and twelve world elite biathlon (Bia) athletes were evaluated for cardiopulmonary exercise test (CPET) performance as the primary aim of our descriptive preliminary report.“

  1. Line 19: Add the word “the”. “...the primary aim of our...”

We want to thank the reviewer for making us aware about this missing word. We amended our revised manuscript on page 1, line 17 to 19:

“Nine Ski mountaineering (Ski-Mo), ten Nordic-Cross Country (NCC) and twelve world elite biathlon (Bia) athletes were evaluated for cardiopulmonary exercise test (CPET) performance as the primary aim of our descriptive preliminary report.“

4. Lines 21-23: the “...and sport specific training aspects.” is not very specific. Should this be defined more clearly?

We want to thank the reviewer for this important comment. The reviewer is absolutely right that the term “aspects” is not very specific. In our revised manuscript we inserted the term “schedule” to define the circumstances more clearly.

page 1, line 21 to 23:

“The matched data of the elite winter sports athletes (14 women, 17 male athletes, age: 18-32 years) were compared for different CPET parameters, and athlete´s physique data and sport specific training schedule.“

5. Line 24: Change “could be” to “were”.

We want to thank the reviewer for making us aware about this inaccuracy. We revised our abstract on page 1, line 24 to 25 to the following:

“Significant differences were revealed…“

6. Line 26-27: Change to “...(VO2/kg VT2), the oxygen pulse at VT2, and the maximum...”

We want to thank the reviewer for making us aware about this inaccuracy. We revised our abstract on page 1, line 27 to the following:

“…VT2 (VO2/kg VT2), the oxygen pulse at VT2, and the maximum oxygen pulse level…”

7. Line 29: Should the phrase “remodeling by CPET analyses” be used here? Did this study detect remodeling or only use CPET data as a measure of physiological adaptation or fitness level? Would it be better to say, “...significant differences in CPET values exist, against the ....”?

Thank you very much for the comment. As stated above, the reviewer is absolutely right that the phrase has to be removed and the fact has to be stated clearly in our revised manuscript.

Therefore the abstract was amended as following, page  1, line 28 to 30:

“This report provides new evidence that in different world elite winter sport professionals significant differences in CPET parameters can be demonstrated, against the background of athlete´s physique as well as training control and frequency.“

Introduction

  1. Lines 36-38: Revise this sentence. Or you could combine it with the first sentence to create a single concise sentence.

We want to thank the reviewer for this important comment. The improve the quality of our revised manuscript we amended the proposed revision of the sentence on page 1, line 35 to 37:

“Ski mountaineering (Ski-Mo), Nordic-Cross Country (NCC), and Biathlon (Bia) are known to be the most challenging winter sports because they involve whole-body movements, uphill locomotion, and altitude environments.”

  1. Here is a possible revision you could use: “Ski mountaineering (Ski-Mo), Nordic-Cross Country (NCC), and Biathlon (Bia) are known to be the most challenging winter sports because they involve whole-body movements, uphill locomotion, and altitude environments.”

Thank you very much for this proposal.

2. Lines 49-51: Revise this sentence.

3. Lines 51-53: change the end of the sentence to “...cardiac imaging, and clinical cardiological assessment.”

We want to thank the reviewer for this important comment. To improve the quality of our revised manuscript we changed this sentence to the following on page 2, line 85 to 86:

“These sport specific remodeling processes are evaluated by CPET analyses, cardiac imaging, and clinical cardiological assessment.”

4. Lines 60-61: You can delete “of these highly endurant athletes.”

5. Lines 65: You can delete “of these high endurant category.”

6. Line 69: Change to “...by utilizing CPET data...”.

7. Lines 68-70: How much information is CPET giving us on cardiac remodeling? Can you elucidate the physiological remodeling from CPET?

We want to thank the reviewer for making us aware of these inaccuracies. The aim of our descriptive preliminary reporting has to be stated more clearly and therefore we amended our revised manuscript on page 2, line 93 to 104 as following:

“In our preliminary report of these professional winter sport athletes, we want to elucidate the exercise induced cardiopulmonary adaption by CPET. In winter sport CPET is known as an important tool to analyze the individual athlete´s performance level and training status [20,21]. In the last decades CPET has emerged as the main important analyzing tool to manage peri-seasons training schedule levels and improve athlete´s performance [4,8,20,22,23,24]. Nevertheless, there is a lack of data about interdisciplinary comparison of world elite winter sport professionals´ CPET data and the individual athlete´s physique´s status and training impact.

Therefore, the aim of our present descriptive preliminary reporting is to reveal physiological performance differences by utilizing CPET data in German elite competitive winter sports athletes.”

Material and Methods

  1. Line 87: What types of movements/activities occurred in the “recreational time”? Should this be defined more clearly?

We want to thank the reviewer for this important comment. The reviewer is absolutely right that these contents have to be defined more clearly in our revised manuscript. Our manuscript was amended on page 2, line 126 to 131:

“All participants were professional athletes with a total amount of 20-25 training hours per week - based on the individual high volume training schedule during the season - and an amount of 5-10 trainings hours of recreational time, including continuous endurance training as cycling and running, functional strength training and individual training to improve muscle dysbalances.“

2. Line 95: Change “AHA” to “American Heart Association (AHA)”

We want to thank the reviewer for this important comment. The sentence was revised on page 3, line 157 to 158:

“The CPET was conducted in accordance to the recommendations of the American Heart Association (AHA)…”

  1. Line 99: change the beginning of this sentence to “Alternatively, the treadmill tests started with a ...”

We want to thank the reviewer for this suggestion to improve the quality of our revised manuscript. The sentence was revised on page 3, line 161 to 162:

“Alternatively, the treadmill tests started with a workload of 8 km/h for 3 min increasing speed by 1 km/h every 3 minutes.“

  1. Lines 100-102: Revise this sentence.

We want to thank the reviewer for this suggestion to improve the quality of our revised manuscript. The sentence was revised on page 3, line 162 to 164:

“We continuously recorded all performance data from the CPET measurements, additionally twelve-lead ECG and blood pressure.“

  1. Lines 102-108: Revise this long sentence.
    1. What specific lactate levels?
    2. What specific exertion levels on the Borg RPE scale? Values ≥ 17?

We would like to thank the reviewer for taking the time to read our manuscript thoroughly and addressing this important point to us. We are aware about this inaccuracy and revised our manuscript on page 3, line 165 to 174 as following:

“To define peak performance criteria for CPET analysis several factors were taken into consideration. First each participant was analyzed for peri- and post-exercise lactate level by capillary blood analysis from the earlobe at each workload step, individual anaerobic threshold (4 mmol/l), and in the recovery time (maximum 15 mmol/l). Furthermore the VO2 max levelling off in the CPET analyses, reaching 85% of the individual maximum predicted heart rate (220 bpm minus age in years), reaching a respiratory exchange rate (RER) 1.15 at peak performance, and reaching individual assumed exercise time of CPET duration were focused in CPET analyses. The specific athlete´s exertion level was analyzed analogue to the Borg RPE scale (Values ≥ 17).“

Results

  1. Line 133: Change “estimated” to “expected”.
  2. Line 134: add “(Table 2)” to the end of the sentence. “...data in the CPET (Table2). Updated Olympic-medal-level...”

We want to thank the reviewer for this suggestion to improve the quality of our revised manuscript. The sentence was changed on page 4, line 282 to 283:

“All athletes showed, as expected in elite winter sport athletes, excellent performance data in the CPET (Table 2).“

  1. Lines 136-140: Delete this paragraph as it does not add anything new to the manuscript.

We would like to thank the reviewer for taking the time to read our manuscript thoroughly and addressing this important point to us. We are aware about this inaccuracy and deleted this paragraph.

  1. Lines 156-158: Should this sentence be in the methods?

We want to thank the reviewer for this suggestion to improve the quality of our revised manuscript. The sentence was added in the 2. Material and Methods section, Measurements, page 3, line 176 to 179:

“The CPET data were analyzed and the peak oxygen pulse (Oxygen pulse maximum) as well as the oxygen pulse at VT2 (Oxygen pulse VT2) were calculated by dividing derived VO2 by the maximum heart rate or the heart rate at VT2 during exercise.”

and the results section was amended as following on page 5, line 425 to 429:

“The Oxygen pulse maximum as well as the Oxygen pulse VT2 were analyzed and the results are presented in detail in Table 2. Focusing this parameter, NCC and Bia athletes showed significant higher Oxygen pulse maximum values compared to Ski-Mo athletes (Figure 2).”

5. Lines 158-160: Delete this sentence: “Focusing this physiological performance parameter, significant differences were elucidated in our participating winter sport professionals.”

We would like to thank the reviewer for taking the time to read our manuscript thoroughly and addressing this important notice to us. The sentence was deleted in the revised manuscript.

6. Lines 172: Put “(results shown in Table 2).” at the end of the first sentence.

7. Lines 172-175: Revise this sentence for conciseness and clarity.

    1. Here is a possible revision you could use: “The peak oxygen pulse (Oxygen pulse maximum) were highest for female Bia compared to female Ski-Mo and NCC.”
    2. You could/should include p-values here too. Also include them in Table 2?

We want to thank the reviewer for these suggestions to improve the quality of our revised manuscript. The sentence was amended as following on page 6, line 457 to 463:

“In this context, analyzing the sex-specific differences of these parameters across the three different disciplines, the female Ski-Mo athletes showed the highest respiratory minute volume (VE) at the second ventilatory threshold (VT2) (p = 0.0286), significant higher performance values for the VO2/kg VT2 in comparison to the female NCC and Bia athletes (p = 0.0286 & 0.0381, results shown in Table 2). The peak oxygen pulse (Oxygen pulse maximum) were highest for female Bia compared to female Ski-Mo and NCC athletes (p = 0.0190).”

8. Lines 181-185: Revise this sentence for conciseness and clarity.

    1. Here is a possible revision you could use: “Additionally, maximum ventilatory oxygen uptake (VO2 maximum) and peak oxygen pulse (Oxygen pulse maximum) were significantly higher for male Bia compared to male Ski- Mo.”
    2. You could/should include p-values here too. Also include them in Table 2?

We would like to thank the reviewer for taking the time to read our manuscript thoroughly and addressing these important notices to us. The sentence was amended as following on page 6, line 465 to page 7, line 486:

“The male athletes differed not significantly with regards to the highest respiratory minute volume (VE) at the second ventilatory threshold (VT2), whereby male Ski-Mo athletes showed the best performance values in comparison to the NCC and Bia athletes (p =0.0519 & 0.0823), but in the end the highest maximum respiratory minute volume (VE maximum) was elucidated for the male Bia athletes, which showed significant differences in comparison to male Ski-Mo athletes (p = 0.0087, results shown in Table 2). Additionally, maximum ventilatory oxygen uptake (VO2 maximum) was significantly higher for male Bia athletes compared to male Ski- Mo (p = 0.0087, Figure 3), no significant differences were revealed for NCC compared to Bia athletes. The significant best performance data referring the peak oxygen pulse (Oxygen pulse maximum) in the three participating male athletes could be proven for the male NCC and Bia athletes (p = 0.0303 & 0.0260, results shown in Table 2).”

Tables

  1. Table 1: The title is in the table. Move it up to the heading: “Table 1: Anthropometric data...”.
  2. Table 1: Do not underline Table heading.

We want to thank the reviewer for these suggestions to improve the quality of Table 1 in our revised manuscript. The Table 1 was modified according to these suggestions and included on page 4.

  1. Table 2: Is there room to include significant p-values for male comparisons and female comparisons?

We want to thank the reviewer for this important suggestion to improve the quality of Table 2 in our revised manuscript. The Table 2 was modified and included on page 5.

Figures

  1. Figure 1: Include the p-value for Ski-Mo vs. NCC (p = .0030) as this is also statistically significant.
  2. Figure 1: Revise the caption to include both significant p-values.
  3. Figure 1 & 3: Delete “Comparison of...” in the figure titles.

We want to thank the reviewer for these suggestions to improve the quality of Figure 1 and Figure 3 in our revised manuscript. The Figures were modified according to these suggestions. Thank you very much for your support.

  1. Figure 3: Where did the p-value come from? I don’t see it anywhere else in the manuscript. Should this significant p- values for male vs. female comparisons be included in Table 2 (see comment #3 in the Table section above).

We want to thank the reviewer for these suggestions to improve the quality of Figure 3 in our revised manuscript. The Figure 3 was modified according to these suggestions. We have to apologize for the inconveniences regarding our inaccuracies. The p-values were added in Table 2 and in the updated Figure 3. Thank you very much for your support

Discussion

  1. Lines 201-203: Change part of the sentence to “...due to the nature of cross-country ski competitions, which involve recovery downhill episodes, high intensity sprints, and uphill racing.”

Thank you very much for the comment on our inaccuracy, we corrected the sentence in the revised manuscript on page 7, line 504 to 505.

2. Lines 203-207: This sentence needs to be revised. I think it can be shorter and still be effective in making your point. For example, “The participants in this study had slightly lower CPET values, defined as VO2/kg maximum, compared to Olympic-medal-level benchmarks from 1990-2013 [27].”

We would like to thank the reviewer for this good comment and amended our revised manuscript on page 7, line 505 to 507 according to the reviewer´s proposal.

3. Lines 209-210: Change “were evaluated for” to “had”

Thank you very much for this notice, our manuscript was changed on page 7, line 510.

4. Lines 216-217: Revise part of this sentence to “...oxygen uptakes, as well as high anaerobic thresholds, although anaerobic thresholds play a less important...”.

Thank you very much for the comment on our inaccuracy, we corrected the sentence in the revised manuscript on page 8, line 656 to 658:

“Even the previous reported physiological demands in NCC athletes requiring high maximal oxygen uptakes, as well as high anaerobic thresholds, although anaerobic thresholds play a less important role, confirm the data of our participating athletes.“

5. Line 219: Change “correlated with” to “and”

Thank you very much for this notice, our manuscript was changed on page 8, line 660.

6. Line 234: Change “parameter” to “parameters”

Thank you very much for this comment, our manuscript was changed on page 8, line 675.

7. Line 236: Use a different word than “pictured”.

  1. Line 237: Change to “...with age, lifetime training hour accumulation, enhanced volume of training, and...”

Thank you very much for these comments, our manuscript was amended as following on page 8, line 676 to 679:

“Our results are even supported by Rusko et al., who revealed an increase of the VO2 maximum in NCC athletes with age, and life time training hour accumulation, enhanced volume of training, and volume of intensive training.”

9. Line 240: Change “could be” to “were”

Thank you very much for this notice, our manuscript was changed on page 8, line 681.

  1. Line 246: Have you defined the LV abbreviation? If not, write out the whole term: “left ventricular (LV)”.

Thank you very much for the comment. We apologize for our inaccuracy and amended our revised manuscript as following on page 8, line 687 to 689:

“Therefore, lower left ventricular (LV) Mass index, less left atrial remodeling as measured by the left atrial (LA) volume index, and lower values for the LV global longitudinal strain (GLS) were proven for the Ski-Mo athletes.“

  1. Line 252: Delete “in the end due to the anatomic and physiologic aspects”

Thank you very much for the comment. We revised our manuscript on page 8, line apologize for our inaccuracy and amended our revised manuscript as following on page 8, line 692 to 695:

“On the basis of this excellent respiratory capacity, the Ski-Mo athletes perform very well at the VT2, but the NCC and Bia athletes represent the physically stronger and more experienced participants to create higher maximum ventilatory oxygen uptake.“

  1. Line 270: Delete “analysis”

Thank you very much for the comment, the term “analysis” was deleted in the revised manuscript on page 9, line 741.

  1. Lines 282-285: This part needs to be revised. Here is a possible revision you could use: “Various studies have investigated the ability of laboratory- and field-based tests to predict elite winter sport athletes’ performance, suggesting that both field-based and laboratory performance indices provide valid prediction of winter sport athletes’ training and race performance [5,24,32,35].
  2. Lines 286-287: Delete this sentence? It does not seem to provide new information.

We would like to thank the reviewer for these important comments and amended our revised manuscript on page 9, line 771 to 774 according to the reviewer´s proposal:

“Various studies have investigated the ability of laboratory- and field-based tests to predict elite winter sport athletes’ performance, suggesting that both field-based and laboratory performance indices provide valid prediction of winter sport athletes’ training and race performance.”

Conclusions

  1. Lines 289-292: Revise this sentence. Can CPET detect physiological remodeling? This is similar to the suggestion to revise the last sentence in the abstract.
  2. Lines 300: Delete “From this aspects” so the sentence starts with “Future studies ...”:

We would like to thank the reviewer for taking the time to read our conclusion thoroughly and addressing these two important notices to us. The conclusion was amended in the revised manuscript on page 9, line 776 to page 10, 823:

“5. Conclusions

This report provides new evidence that in different world elite winter sport professionals significant differences in CPET parameters can be demonstrated, against the background of athlete´s physique and anthropometric data as well as training control and frequency.

Our results might serve as a preliminary report and have to be handled with care due to the mentioned limitations. Therefore our results – in general as well as in the gender specific subgroup analyses – can reveal sport specific individual differences in the CPET performance of our three participating world elite winter sport professionals, especially due to the VE VT2, VE maximum, VO2/kg VT2, Oxygen pulse VT2, and Oxygen pulse maximum. These differences between the groups might be defined by the athlete´s individual respiratory capacity as well as the functional and structural sport specific cardiac remodeling. Nevertheless interpreting the athlete´s individual performance in CPET the impact of athlete´s physique, the sport specific cardiac remodeling, the different main emphasis in training schedule, and accumulation of life time training hours have to be taken into consideration.

Future studies with a greater number of participating winter sport professionals and long term follow up periods have to be considered to elucidate these findings and strengthen the scientific evidence base.”

Abbreviations

  1. Lines 327-332: Include CPET

We have to apologize for the inaccuracy. The abbreviation was added in our revised manuscript on page 10, line 863. Thank you very much for your support.

References

  1. Article titles – Some of them have only the first letter of the first word capitalized but not the remaining words in the title vs. others have the first letter of each word capitalized. Make it consistent throughout the reference list.
  2. Reference #6 = Author names L.A., and S.L.??
  3. Reference #6 = The journal name should be written out instead of abbreviated.
  4. Reference #37 = The journal name should be italicized.

We have to apologize for the inconveniences regarding our inaccuracies in the reference section. In the revised manuscript we provide a consistent reference list and corrected these important points of the reviewer. Thank you very much for your support to improve the quality of our manuscript.

Reviewer 3 Report

The abstract of this article was clear and seemed consistent with the contents.  It appears that the hypothesis is clearly stated and since it is a new approach, the methodology seemed appropriate and clearly stated.  References were clearly stated and the degree with which they were used was clear.

it would be important to review the grammar, and English formatting in the discussion.

With respect to specifics in the manuscript:

  • The manuscript is clear, and relevant to the  likely readers and it definitely was well structured and readable.  Especially noteworthy would be the uniqueness of the inquiry.
  • References were clear and did not seem to have too many self-citations.
  • The design of the study was/is appropriate and is clearly noted in the information about the purpose, procedure, results and discussion of the manuscript.
  • the tables, figures , and notations were clear, readable and supported the narrative.
  • the coordination of the methods, reasoning, results, discussion and purpose was effective.
  • appropriate information was available with respect to consent for inclusion in the study.
  • Especially of note is the consideration, though with small numbers. the information related to male, vs female, genetics  and age considerations along with training protocols.

Author Response

Reviewer 3:

Dear Editor,

Dear Reviewers,

Thank you very much for reviewing our manuscript entitled “Physiological aspects of world elite competitive German winter sport athletes”. Please find below a point-to-point response to the specific comments.

Comments and Suggestions for Authors

The abstract of this article was clear and seemed consistent with the contents.  It appears that the hypothesis is clearly stated and since it is a new approach, the methodology seemed appropriate and clearly stated.  References were clearly stated and the degree with which they were used was clear.

It would be important to review the grammar, and English formatting in the discussion.

We highly appreciate the willingness to review our manuscript and also express our thanks for the positive feedback.

Additional thanks for the positive comments and suggestions for the authors.

We would like to thank the reviewer for addressing this important point to review the grammar and English formatting. We are aware about this inaccuracy and revised our manuscript by an English native.

With respect to specifics in the manuscript:

  • The manuscript is clear, and relevant to the  likely readers and it definitely was well structured and readable.  Especially noteworthy would be the uniqueness of the inquiry.
  • References were clear and did not seem to have too many self-citations.
  • The design of the study was/is appropriate and is clearly noted in the information about the purpose, procedure, results and discussion of the manuscript.
  • the tables, figures , and notations were clear, readable and supported the narrative.
  • the coordination of the methods, reasoning, results, discussion and purpose was effective.
  • appropriate information was available with respect to consent for inclusion in the study.
  • Especially of note is the consideration, though with small numbers, the information related to male, vs female, genetics  and age considerations along with training protocols.

We would like to thank the reviewer for taking the time to read our manuscript thoroughly and addressing these positive comments on the manuscript.

Round 2

Reviewer 1 Report

The authors have edited the text in accordance with the comments, I have no further comments. Congratulations to the authors on an interesting study.